# Nutritional Content of Non-Dairy Frozen Desserts

**DOI:** 10.3390/nu14194150

**Published:** 2022-10-06

**Authors:** Winston J. Craig, Cecilia J. Brothers

**Affiliations:** 1Center for Nutrition, Healthy Lifestyles and Disease Prevention, School of Public Health, Loma Linda University, Loma Linda, CA 92354, USA; 2Department of Biology, Walla Walla University, College Place, WA 99324, USA

**Keywords:** plant-based frozen desserts, vegan ice cream, sugar, fat, saturated fat, iron, protein

## Abstract

There is a growing interest in non-dairy alternatives fueled by concerns about personal health and the health of the planet. Sales of non-dairy frozen desserts have increased along with other non-dairy alternatives such as plant-based beverages, cheeses, yogurts and creamers. The aim of this study was to conduct a cross-sectional survey of plant-based frozen desserts to determine their nutritional content. A total of 358 plant-based frozen desserts were analyzed from the nutrition label listed on the commercial container. The various products were based upon coconut (*n* = 126), oat milk (*n* = 63), almonds (*n* = 42), cashews (*n* = 25), soy (*n* = 11), macadamia milk (*n* = 9), olive oil (*n* = 8), faba bean (*n* = 8), canola oil (*n* = 8), rice milk (*n* = 6), sunflower milk (*n* = 6), avocado (*n* = 5), pea protein (*n* = 5) and various fruits, nuts and mixed blends (*n* = 36). While 90% of the frozen desserts had high sugar levels, 73% had high levels of saturated fat (due to the presence of coconut oil) and only one in four had high levels of fat. None of the products were fortified with calcium, vitamin D or B12, but one in six products had iron levels/serving of at least 10% of Daily Value (DV) and 1 in 6 had protein levels/serving similar to regular dairy ice cream. Food manufacturers need to produce new non-dairy frozen desserts that are more nutritious, since few brands (such as those based upon avocado, apple and hemp protein, or fava bean) presently provide consumers choices with lower saturated fat and sugar levels and/or higher protein levels.

## 1. Introduction

Flavored ice has been utilized for centuries, first by the Chinese and then the Persians. It was during the 17th century that sweetened sorbet (sorbetto) was created by the Italian Antonio Latini and frozen desserts grew in popularity. He is credited with creating a milk-based sorbet, which culinary historians consider the first official ice cream [1,2]. By the 18th century, ice cream was very popular in Europe [1]. Today, ice cream is one of America’s favorite desserts. Americans consume an average of 23 gallons of ice cream each year, with vanilla and chocolate the most popular flavors [1]. About 9% of American cow milk production is devoted to making ice cream [2]. In the US, ice cream must contain at least 10% milkfat according to Food and Drug Administration (FDA) regulations [3].

With growing concerns about sustainable diets, planetary and personal health issues [4], an increasing number of consumers are choosing plant-based foods including milk, yogurt and cheese alternatives, creamers and frozen desserts made from a variety of grains, legumes, nuts, seeds and fruits. The vegan non-dairy ice cream global market was valued at $521 million in 2019 [5] and is expected to grow at a substantial rate over the next decade. Europe was the greatest contributor to the market with a 36% share in 2019 [5]. The gradual shift of consumers from dairy-based to non-dairy-based frozen products is propelling the product demand, especially among younger generations and flexitarians [6,7].

In our study, we set out to identify the variety of non-dairy frozen desserts available in the US marketplace and to document their nutritional composition. The focus centered especially on fat, saturated fat, sugar and caloric content since desserts are often classified as healthy or otherwise based on the level of these nutrients. The importance of the proposed study was highlighted by the dearth of nutritional information available for non-dairy frozen desserts, at a time when such products are growing in popularity. This appears to be the first comprehensive nutritional study of non-dairy frozen desserts.

## 2. Materials and Methods

The nutritional content of 358 plant-based, non-dairy frozen desserts was analyzed. In addition, 55 frozen non-dairy bars and sandwiches, representing 14 brands, were analyzed for nutritional content. The frozen desserts were most commonly found as pint containers. Two brands containing an animal-free milk protein made from a genetically engineered fungi were excluded. Sorbets, smoothies, sundaes, cones, scoops, frozen custards and frozen yogurts were also excluded from this project. Only the varieties with complete nutritional data were included in the analysis. The non-dairy frozen desserts, bars and sandwiches were located in supermarkets in the western United States during the period January to April 2022. Furthermore, additional varieties were located on the website of the manufacturer or online commercial retailer. The nutrients per serving size, were recorded from the nutrition labels available on all packages or websites and included calories, fat, saturated fat, sodium, carbohydrates, dietary fiber, total sugars, protein, calcium and iron. Median values of the nutrients were calculated for each frozen dessert, bar or sandwich. Further details on data collection of product samples appears in the Supplement Section S1.

The nutritional value of each plant-based non-dairy frozen dessert was rated according to the criteria used in our earlier non-dairy publications [8,9,10]. The US Dietary Guidelines were used as a guide to specify a low or high nutrient value. A nutrient/serving is considered low when it is present at 5% of DV or less, while a nutrient/serving is considered high when it exists at 20% DV or more [11,12]. In the USA, the DV for sodium is 2300 mg, protein is 50 g, saturated fat is 20 g, added sugars is 50 g, fat is 78 g, iron is 18 mg, while the daily intake is based upon a 2000 calorie diet [12]. Ten percent was chosen as a mid-stream number between the 5% DV (low value) and the 20% DV (high value). This 10% of DV level gave us a minimally acceptable level of 5 g protein/serving and 1.8 mg for iron/serving, while levels of not more than 5 g sugars/serving and 200 calories/serving were considered acceptable levels. For low levels of sodium and saturated fat, this translated into not more than 115 mg sodium (5% of DV) and not more than 1 g saturated fat (5% of DV). We also wished to measure the frequency of products with high levels of fat, saturated fat and sugar. This was recorded as the number of frozen desserts having more than 15.5 g fat/serving (20% DV for fat), having 4 g or more of saturated fat/serving (20% DV) and having 10 g or more of total sugars/serving (20% DV).

Sweetening agents other than the usual cane sugar, corn syrup and tapioca syrup were noted. Added oils other than the commonly used coconut oil were documented, as well as the presence of any prebiotics or probiotics and the variety of gums used.

### Statistical Analysis

R software was used to conduct all statistical analyses [13]. Data were tested for normality and homoscedasticity prior to analysis. The median and interquartile range were used for descriptive statistics, as the data were not normally distributed. The nutritional content was compared across the types of non-dairy frozen dessert bases using a Kruskal-Wallis test for each nutrient, followed by Dunn’s post hoc test with Bonferroni adjustment for pairwise comparisons between base types. A significant *p*-value of less than 0.05 was used for all analyses.

## 3. Results

The 358 plant-based frozen desserts, representing 48 brands, that were analyzed were based upon coconut (*n* = 126), oat milk (*n* = 63), almond (*n* = 42), cashews (*n* = 25), soy (*n* = 11), macadamia milk (*n* = 9), olive oil (*n* = 8), faba bean (*n* = 8), canola oil (*n* = 8), rice milk (*n* = 6), sunflower milk (*n* = 6), avocado (*n* = 5), pea protein (ripptein) (*n* = 5), acai (*n* = 3), banana (*n* = 3), cocoa (*n* = 1), peanuts (*n* = 1) and the following mixtures: apple puree and hemp seeds (*n* = 9), pear puree and nuts (*n* = 7), oats and hemp milk (*n* = 6), banana and coconut (*n* = 3), tapioca and rice flour (*n* = 1), almonds and sunflower oil (*n* = 1), wheat and palm oil (*n* = 1). Table 1 displays the medians of each nutrient for all base types and the differences among the base types are reported. All nutrients varied significantly across base type (Appendix A).

The nutritional quality of the frozen desserts varied widely depending upon the base type. Eight products (one brand) contained no fat, while 24% of the desserts had high levels of fat. Coconut oil was by far the predominant oil present. All but seven brands contained coconut oil. Palm oil was present in 8% of the products. Other major oils used included sunflower oil (6%), canola oil (5%), peanut oil (4%), avocado oil, palm kernel oil and soy oil (each 3%). Corn, safflower and rice bran oils occurred in less than 2% of the frozen desserts.

While almost 9% of the products had no added sugar, 90% had high levels of sugar. A variety of other sweetening agents were used among the various brands. Ten products (three brands) used dates while five products (one brand) used honey to sweeten. Thirty-four products (9.5%) used agave syrup, 30 products (8.4%) contained the sugar alcohol erythritol, 28 (7.8%) contained monk fruit, 16 (4.5%) used allulose, 15 (4.2%) sweetened with stevia, while just two products used lucuma powder.

Pea protein was a common ingredient, being identified in 31% of the pint products. Prebiotics were more commonly an ingredient than probiotics. Inulin or chicory root extract was present in 73 (20.4%) of the products. Just one brand (eight varieties) added seven active cultures to their product.

Carob bean gum and guar gum were commonly used as thickening agents and stabilizers. Other gums less commonly used were xanthan gum and gum acacia, while a few contained tara gum, carrageenan and tamarind. None of the frozen desserts were fortified with either vitamin D, vitamin B12 or calcium. Seventy-six of the desserts contained chocolate, while 44 were flavored with vanilla, 42 had cookie dough, 38 had caramel, 29 had mint flavor, 25 contained peanut butter, 20 had strawberry flavor and 19 contained coffee.

Table 2 summarizes the data showing the percentage of the frozen desserts that contain (a) low levels of sodium and saturated fat and modest levels of calories and sugar, (b) a reasonable level of protein and iron and (c) high levels of sugar, fat and saturated fat.

Fifty-five frozen non-dairy bars and sandwiches, representing 14 brands were analyzed. These products were based upon coconut (*n* = 23), oats (*n* = 15), almond (*n* = 7), cashew (*n* = 4), palm oil/soy (*n* = 3), banana (*n* = 2), sunflower (*n* = 1). The median for one serving (a sandwich or chocolate-covered bar) was 64 g. Table 3 displays the medians of each nutrient for all base types for the frozen bars and sandwiches and the differences among the base types are reported. Only saturated fat and sodium content varied significantly across base type (Appendix A).

## 4. Discussion

Of the 358 varieties analyzed, representing 48 brands, we discovered 24 different base types with coconut, oats, almonds and cashews being the most common base type. The huge variety of plant-based frozen desserts gives the consumer a variety of choices. Most consumers look for an ice cream, or a non-dairy alternative that is creamy and sweet, without an excessive number of calories. In this study, two in five products had 200 calories or less (10% of daily value or DV). Increasingly, consumers want healthier products, without too much saturated fat, fat and sugar, nutrients often associated with chronic diseases [14,15,16]. Sugar provides not only sweetness but also texture to the product.

As expected, most of the frozen desserts were high in sugar. About 90% of the products had 10 g (20% of DV) or more of sugar per serving. Due to the presence of coconut oil in many of the products, over 70% had high levels of saturated fat. On the other hand, only 24% of the desserts had high fat levels (over 15.5 g of fat or 20% of DV). About 70% of the desserts had low levels of sodium (no more than 115 mg sodium/serving or 5% of DV). Non-dairy frozen desserts are available that provide lower saturated fat and sugar contents.

From Table 1 we see that coconut-based products have significantly more saturated fat than nine other based products (almond, apple/hemp, avocado, canola oil, faba protein, oats/hemp, olive oil, rice and soy). Furthermore, the apple/hemp-, avocado- and faba bean-based products contain significantly less sugar than almond-, canola oil-, olive oil-, rice- and sunflower-based products.

Some companies make an effort to provide low sugar options for their dairy-free frozen desserts [17]. These products may contain one or more of the following low-calorie sweeteners as a sugar alternative: erythritol, allulose, monk fruit extract and/or stevia. These substances have little, if any, caloric value and do not raise the blood sugar levels making them useful for persons with diabetes and those wanting to manage their weight [18,19]. Erythritol a sugar alcohol, has 70% of the sweetness of sucrose but only 5% of the caloric value [20] and may cause gastrointestinal discomfort and diarrhea in large quantities [20]. Allulose, an isomer of fructose, has antioxidant activity and has 70% of the sweetness of sucrose. In addition, D-allulose shows anti-hyperglycemic effect in people with glucose intolerance [21]. Monk fruit (or Luo han guo) extract is up to 250 times sweeter than sugar, due mainly to the presence of compound mogroside V [22]. Monk fruit extract has no energy value and has exhibited anti-hyperglycemic effects [20]. A triterpenoid glycoside from monk fruit has been shown to have anti-proliferative activity with the potential for treating colorectal cancer [23]. Stevia extract contains glycosides that are zero calorie sweeteners and are 200–400 times sweeter than sucrose. All these alternative sweeteners have GRAS (General Recognized As Safe) status as designated by the FDA in the US [20].

One brand of frozen dessert used honey for sweetening while three brands sweetened with dates and five brands used agave syrup. These sweeteners are considered healthy alternatives to sugar [19,24].

Coconut oil was used in 41 of the 48 brands (85%) of frozen desserts. Because of its high saturated fat content, the coconut oil influenced the median level of saturated fat for the combined dairy-free frozen desserts. The overall median was 8 g of saturated fat with a range for those with coconut oil of 7–13 g saturated fat. Those products containing other plant oils such as olive, soy, canola, or avocado, but not containing coconut oil, had a median value that ranged from 0 to 3 g of saturated fat. Almost 11% of the desserts had low saturated fat levels (Table 2). The saturated fat in coconut oil is considered to have an unhealthy impact on cardiovascular health [25,26]. While coconut oil (and especially virgin coconut oil) produces a better lipid profile than the animal fats, other plant oils are considered a better choice for one’s cardiovascular health [27].

Most of the base types (11 out of the 17 base types in Table 1) were low (below 5% DV) in protein. The desserts that had three or more grams of protein were typically those based upon or containing nuts and seeds, rather than the expected ones based upon pea protein or soy. While soy- and pea protein-based non-dairy beverages and yogurts were observed to contain the highest protein levels/serving among the many base types [8], such was not the case for the frozen desserts. Soy- and pea protein-based frozen desserts averaged only 1.5 and 2 g/serving, respectively, lower than six (35%) other base types. It is interesting to note that soy-based beverages were far more common (17%) among the beverage base types than was soy-based frozen desserts among the base types available (3.1%).

About 15% of the products (a total of 53) contained at least 4 g of protein, a value commonly given for the protein level of regular dairy ice cream [28]. All five brands that had the highest protein level (3 g and above) contained nuts or seeds, but the presence of nuts and seeds was not a guarantee of a high protein level, since the protein level of macadamia nut-based products was only 1 g and the sunflower seed based desserts had an average of 2 g protein. The frozen desserts based upon fruits (apple or pear) had the highest protein levels, as the food companies had increased the protein level by adding nuts or seeds.

In a comparison of 45 dairy ice creams (13 brands) and 14 dairy-free frozen desserts (eight brands) by the Center for Science in the Public Interest (CSPI), the dairy-free alternatives were observed to have a higher level of saturated fat and added sugars/serving, but only one-half the median protein level/serving than the dairy options [29]. The data are summarized in Appendix A. The common use of coconut oil in most of the dairy-free products explains their high saturated fat content. The median values for calories, saturated fat, sugar and protein (Appendix A) match very closely with the medians for those nutrients given in Table 1 for our results.

None of the frozen desserts were fortified with calcium, or with vitamin D or B12, in contrast to what we found in non-dairy beverages and yogurts [8,10,30]. Regular dairy ice creams typically provide about 110–130 mg/serving or about 10% DV of calcium and negligible amounts (zero to 4% DV/serving) of vitamin D [28,31,32]. Vitamin B12 levels in ice cream typically range from 0.2 to 0.5 mcg/serving (8 to 20% DV) [33,34]. A significant number of frozen desserts (16%) had at least 10% of the DV/serving for iron. This was typical for those products containing peanut butter. While the non-dairy products contain no cholesterol, the regular dairy ice creams contain about 40–70 mg cholesterol/serving [28,31,32].

One brand of frozen desserts contained seven live cultures, similar to those added to non-dairy yogurts [8]. About 30% of the frozen desserts contained inulin, or the inulin-containing products chicory root or agave syrup. Inulin consumption is reported to enhance the absorption of minerals, supports a healthy immune system, improves bowel function and helps manage blood sugar levels [35]. Both prebiotic (inulin extracts) and probiotic (live cultures) substances modulate human colonic flora and maintain a healthy gut function that facilitates the health and well-being of the consumer [36,37]. Carob bean gum and guar gum were the most commonly used substances as stabilizers in the frozen desserts. These water-soluble gums are reported to be useful for managing glycemia and hypercholesterolemia [38,39]. Eleven products were identified as soy-based (Table 1). Soy is reported to provide a variety of useful health benefits for the consumer [40].

A smaller sample of frozen desserts in the form of chocolate-coated bars and sandwiches was compared with the 358 pints. Comparing Table 1 and Table 3, we see that the nutrient levels/serving (except protein) of the bars and sandwiches are 50–75% of the nutrient levels/serving of the pints. Given that the median serving size of the bars and sandwiches is about 50–60% that of the pints, it is apparent that the smaller amounts of calories, fat, saturated fat and sugar in the bars and sandwiches is due to their smaller serving size. The type of base did not affect the calorie, fat, carbohydrate, sugar or protein content (Table 3). This may be due to the small sample size and the variable serving size of the frozen desserts.

Consumers who desire a more healthful or nutritious non-dairy frozen dessert have very few choices available. Table 4 summarizes the data showing the base types that have the more healthful nutrient levels/serving. Included in the table are the brands currently available that most importantly have lower levels of saturated fat and sugar and a higher level of protein comparable to the dairy product. Apple/hemp protein-based desserts have all three of those above-mentioned characteristics, while avocado and fava bean-based desserts both have two of those three characteristics. All three brands avoid coconut oil and two of the brands enhance their sweetness by use of monk fruit extract and/or fruit puree. These desserts would be considered the healthiest choices, when consumed in moderation. Consumers should be encouraged to read nutrition labels to educate themselves so they can make appropriate choices for better nutrition.

Manufacturers need to provide healthier and more nutritious non-dairy frozen dessert options for those consumers who wish to consume more than just a tasty, sweet dessert. For a start, this could be achieved using lower levels of coconut oil and sugar and a more frequent use of fruit puree. Food manufacturers should also consider fortifying these desserts with calcium to levels similar to those found in regular dairy ice-cream, namely 10% of the DV per serving [28].

For the pints, the frozen dessert with the lowest level of calories, fat, saturated fat and sugar/serving was the fava bean-based product. For the bars and sandwiches format, those made with almond milk showed the lowest median level for calories and fat, a low level of saturated fat and the highest median level of protein and fiber.

Finally, in addition to the various nutritional differences between dairy ice-cream and a plant-based alternative, the selection of a non-dairy frozen dessert would be considered a favorable step to lowering greenhouse gas emissions, as well as lowering land and water use, thereby impacting human-induced climate change favorably [41,42].

The strength of this study is that this is the first reported comprehensive and detailed nutritional analysis of non-dairy frozen desserts available in the US. A limitation of this study is that the data reported here represents just a window on the marketplace for non- dairy frozen desserts in early 2022. Products are being discontinued and new ones are being introduced all the time. Furthermore, the nutritional composition of the desserts was determined only from the nutrition label provided by the manufacturer and no chemical analyses were performed. If an inconsistency was found in the data between the manufacturer’s web site nutritional label information and the information as it appears on the container found in the supermarket, the authors utilized the latter.

## 5. Conclusions

The popularity of non-dairy frozen desserts continues to rise. New varieties, flavors and brands are emerging constantly. A total of 358 plant-based frozen desserts representing 48 brands and based upon a nut, seed, grain, legume, vegetable oil, fruit or a blend of these plant foods, were analyzed. While 35% of the products were coconut oil-based, a total of 41 of the 48 brands contained some coconut oil or cream. This resulted in 73% of the products having high saturated fat levels. While 90% of the frozen desserts had high sugar levels, only one in four had high levels of fat. None of the products were fortified with calcium, vitamin D or B12 in contrast to what the authors reported earlier for non-dairy beverages and yogurts. One in six products had iron levels/serving of at least 10% of Daily Value (DV) and one in six had protein levels/serving similar to dairy ice cream. The non-dairy frozen desserts also have the advantage of being cholesterol free.

While dairy ice cream is a sweet dessert, nutritionists do not consider it to be a major dietary source of calcium, protein or vitamin D or B12. Its high level of sugar and saturated fat can be a concern if used commonly and in large quantities. While the non-dairy alternatives commonly have high levels of sugar and saturated fat, there are a few better choices (different varieties) available that have lower levels of saturated fat and sugar and a level of protein comparable to the dairy product. These include the apple/hemp protein-based desserts in addition to those based upon avocado and fava bean. Some plant-based varieties also contain small amounts of iron and soluble fiber that enhance their nutritional value.

Food manufacturers need to provide healthier and more nutritious non-dairy frozen dessert options for consumers, by finding alternative sources of fat and sweetness beyond coconut oil and sucrose. More products need to be available to have protein levels of 5–7 g of protein per serving, as we found with the apple/hemp protein-based and pear and nuts-based desserts. In addition, it would be a nutritional advantage to have the food manufacturers fortify their frozen desserts with calcium to a level of 10% DV per serving, similar to that of regular dairy ice-cream. Our results provide a guide for consumers to make better choices and to influence food companies in their future formulation decisions.

## Figures and Tables

**Table 1 nutrients-14-04150-t001:** The median (Q1–Q3) values of calories and 7 nutrients of non-dairy frozen desserts/serving classified according to their bases. Data are listed for pint containers only.

Base	*n*	Calories	Fat (g)	Sat. Fat (g)	Sodium (mg)	Carbohydrates (g)	Fiber (g)	Sugars (g)	Protein (g)
Almond	42	250 (202.5–320) ^abcde^	12 (9.3–16) ^abc^	7 (6–9.8) ^abcd^	97.5 (76.3–130) ^ab^	32.5 (27.3–40.8) ^ab^	1 (0–2) ^ab^	23 (0–2) ^ab^	3 (2–3.8) ^abc^
Apple/hemp	9	170 (170–180) ^abf^	6 (5–6) ^a^	1.5 (1.5–1.5) ^ae^	210 (200–220) ^ab^	28 (28–28) ^acde^	8 (7–9) ^c^	14 (14–15) ^cd^	7 (7–7) ^a^
Avocado	5	170 (170–170) ^abf^	12 (11–12) ^abcd^	1.5 (1.5–1.5) ^abce^	45 (45–45) ^acd^	18 (18–18) ^cd^	1 (1–1) ^abcd^	12 (12–13) ^cd^	1 (1–1) ^de^
Canola oil	8	325 (295–340) ^c^	20 (17.3–20.5) ^b^	2 (1–2.6) ^ae^	230 (215–267.5) ^b^	33 (30.3–36.3) ^abe^	1 (0–1.3) ^abd^	24 (22.8–28.3) ^ab^	3.5 (3–4) ^abcf^
Cashews	25	240 (174–260) ^abcde^	13 (7–16) ^abc^	9 (1–11) ^abcdf^	125 (64–170) ^abc^	29 (19–33) ^ace^	1 (0–2) ^ab^	23 (17–25) ^abc^	3 (2–3) ^abcdf^
Coconut	126	210 (180–250) ^ad^	12 (9–16) ^bc^	10 (8–13) ^f^	53 (15–90) ^cd^	24 (21.3–28) ^cde^	3 (1–6) ^cd^	17 (13–21) ^d^	2 (1–2) ^de^
Faba bean	8	50 (50–50) ^f^	0 (0–0) ^d^	0 (0–0) ^e^	90 (90–90) ^abcd^	11 (11–11) ^d^	3 (3–3) ^acd^	8 (8–8) ^d^	2 (2–2) ^abcdef^
Macadamia	9	202 (197–237) ^abcdef^	10 (9–11) ^acd^	7 (6–7) ^abcdef^	25 (24–33) ^c^	29 (29–31) ^abce^	1 (0–1) ^abd^	18 (17–20) ^abcd^	1 (0–1) ^e^
Oats	63	240 (210–260) ^bce^	11 (9.5–15) ^abcd^	9 (6.5–11) ^bdf^	65 (42.5–125) ^acd^	33 (30–36) ^abe^	1 (0–2) ^ab^	20 (18–22) ^abc^	2 (1–2) ^def^
Oats/hemp	6	215 (202.5–220) ^abcdef^	12 (11.3–12) ^abcd^	2 (2–3.1) ^abce^	87.5 (85–93.8) ^abcd^	28 (24.8–29.8) ^abcde^	5 (2–5.8) ^abcd^	20 (17.5–21.8) ^abcd^	1 (1–2.5) ^bcdef^
Olive oil	8	190 (190–202.5) ^abdef^	8 (7–9.3) ^acd^	1 (1–1.5) ^ae^	90 (88.8–197.5) ^abcd^	29 (28–30.3) ^abce^	0 (0–0) ^b^	24.5 (23.8–25) ^ab^	1 (1–2.3) ^bdef^
Pea protein	5	260 (250–270) ^abcde^	15 (14–15) ^abc^	13 (13–14) ^df^	100 (90–105) ^abcd^	31 (29–31) ^abce^	1 (1–1) ^abcd^	21 (20–21) ^abcd^	2 (2–2) ^abcdef^
Pear & nuts	7	280 (270–290) ^cde^	15 (13.5–15.5) ^abc^	4.5 (4.5–5.5) ^abcdef^	150 (145–200) ^ab^	33 (31–33) ^abce^	2 (1.5–3) ^abcd^	19 (19–20) ^abcd^	6 (5–6) ^ac^
Rice	6	255 (235–267.5) ^abcde^	13.5 (12.25–14.8) ^abc^	3 (2.25–4.6) ^abce^	138 (123.8–155) ^abd^	33.5 (29.8–35) ^abce^	1 (0.25–1) ^abd^	26.5 (24.3–28) ^ab^	2 (1.3–2) ^abcef^
Soy	11	240 (180–250) ^abcdef^	12 (7–14.5) ^abcd^	3 (2–3.5) ^ae^	95 (85–105) ^abcd^	26 (23.5–31.5) ^abcde^	4 (1–4) ^acd^	17 (14–25.8) ^acd^	1.5 (1–3) ^bdef^
Sunflower	6	330 (302.5–350) ^c^	14 (14–17) ^bc^	10 (10–10.8) ^bcdf^	133 (121.3–155) ^abcd^	45 (41–48.3) ^b^	1 (0–0.8) ^abd^	28.5 (27.3–30.5) ^b^	2 (2–2) ^abcdef^
Others	14	210 (142.5–312.5) ^abcde^	6 (4–11.8) ^acd^	3 (2.625–3.5) ^ace^	30 (28–163.8) ^abcd^	34.5 (27.3–46.3) ^abce^	3 (1–4) ^acd^	24.5 (24–25.8) ^ab^	2.5 (1–3) ^abcdf^
Total	358	230	12	8	80	29	2	19	2

Kruskal–Wallis non-parametric tests were used for each nutrient to perform comparisons among base types. The “total” category was not included in the analysis. Dunn’s test of multiple comparisons with Bonferroni adjustments were used to perform pairwise comparisons. Different lowercase letters (a, b, c, etc.) in the same column indicate significant post-hoc pairwise comparisons between ice cream bases. DV is the daily value.

**Table 2 nutrients-14-04150-t002:** Percentage of 358 non-dairy frozen desserts as pints meeting or exceeding suggested guideline/serving.

Not More than	*n*	
200 calories (10% DV)	137	38.3%
115 mg sodium (5% of DV)	250	69.8%
1 g saturated fat (5% of DV)	38	10.6%
5 g sugar (10% DV)	18	5.0%
At Least		
4 g protein	53	14.8%
5 g protein (10% of DV)	34	9.5%
1.8 mg Iron (10% of DV)	57	15.9%
High Levels		
10 g or more sugar (20% of DV)	323	90.2%
4 g or more saturated fat (20% of DV)	260	72.6%
Over 15.5 g fat (20% of DV)	86	24.0%

**Table 3 nutrients-14-04150-t003:** The median (Q1–Q3) values of calories and 7 nutrients of non-dairy frozen bars and sandwiches/serving classified according to their bases.

Base	*n*	Calories	Fat (g)	Sat. Fat (g)	Sodium (mg)	Carbohydrates (g)	Fiber (g)	Sugars (g)	Protein (g)
Almond	7	100 (95–205)	4.5 (3.5–10)	3 (1.5–7) ^ab^	50 (30–100)	21 (17–25.5)	3 (1–5.5)	7 (5.5–14.5)	3 (1.5–3)
Cashew	4	190 (177.5–212.5)	14 (13–15.35)	10.5 (9–12) ^a^	47.5 (43.8–62.5)	16.5 (15.75–19)	1.5 (0–1.3)	13 (12.8–15.3)	1 (1–1.3)
Coconut	23	170 (110–205)	12 (6–14)	10 (5–11) ^a^	30 (7.5–40)	17 (14.5–23.5)	2 (1–3)	13 (10–15)	1 (1–4)
Oats	15	165 (120–230)	7 (5–15.5)	5 (4.5–10) ^ab^	45 (25–65)	19 (17.5–25)	1 (0.8–1)	13 (12–17)	2 (1–2)
Others	6	125 (120–130)	6 (6–6)	2.25 (1.3–4.4) ^b^	97.5 (32.5–118.3)	17 (16.3–18.5)	1 (0.3–1)	12.5 (9.5–14)	2 (0.3–2)
Total	55	170	7	6	40	18	1	13	2

Kruskal–Wallis non-parametric tests were used for each nutrient to perform comparisons among base types (Appendix A). The “total” category was not included in the analysis. Dunn’s test of multiple comparisons with Bonferroni adjustments were used to perform pairwise comparisons. Different lowercase letters (a, b) in the same column indicate significant post-hoc pairwise comparisons between ice cream bases.

**Table 4 nutrients-14-04150-t004:** Frozen Desserts with More Healthful Nutrient Levels Shown by Base Type for Pints Only.

Nutrient/Serving	Apple & Hemp Protein	Avocado	Fava Bean	Oat & Hemp Milk	Olive Oil	Pear Juice & Nuts	Soy
Less than 200 calories	√	√	√		√		
Less than 10 g fat	√		√		√		
Less than 2 g saturated fat	√	√	√		√		
4 g or more of fiber	√			√			√
Less than 15 g of sugar	√	√	√				
4 g or more of protein	√					√	

## Data Availability

Not applicable.

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
