# Peer review of "Nutritional Content of Non-Dairy Frozen Desserts"

_nutrients, 2022, doi:10.3390/nu14194150_

Round 1

Reviewer 1 Report

General comments

  • Overall, the paper shows a thorough look into the availability and nutrient contents on non-dairy frozen desserts in the US market. The paper is overall well written. However there is a concern on the significance of the content and the practical implications of using these findings. This is due to the focus on the nutrient content of non-dairy frozen desserts alone being too restrictive, when there are many other plant based categories to consider with, for analysis. The paper could benefit from a more in-depth discussion on the scientific/practical significance of the results. 
  • It would have been interesting to compare the nutritional values to existing recognized profiling systems
  • The review is missing practical guidance: are some bases more recommended/more nutritious than others? This could be discussed more in depth in the discussion
  • What could be the reasons for such a broad variety in nutritional quality? Can the authors make some hypotheses? (brand, product base, target consumer group, …)
  • Are there any claims made on these products, and if so, are those with claims of better nutritional quality?

Materials and methods

The methodology of data collection needs to be further described. Lines 57-58 – Which or how many supermarkets were visited? What was the methodology of data collection and how was data collected to include or exclude products selection? Who collected the data?

Results

Table 1 was not easy to read. The lower case letters used to indicate significance was confusing as each and every value in the table had letters affixed. It would be better to use either an * or highlight significant pairs. 

Table 5 shows healthful nutrient levels for 1 brand for each base type, how was this 1 brand selected compared to others?

 Discussion

The authors mentioned dairy desserts in the introduction and conclusion as being different from plant based desserts. However, the manuscript lacked a comparison analysis of the non-dairy desserts with dairy products, which could be added to the discussion since the authors concluded plant based desserts to have high sugar and saturated fat too.  

References

There is concern 12 references out of 38 were from websites that may not be scientifically credible.

References questioned:

1. Yuko, E. The History of Ice Cream, One of the World’s Oldest Desserts. March 2022. https://www.rd.com/article/who-invented- 269 ice-cream/. Accessed 2 August, 2022. 270

2. Avey, T. Explore the Delicious History of Ice Cream. July 2012. https://www.pbs.org/food/the-history-kitchen/explore-the- 271 delicious-history-of-ice-cream/. Accessed 2 August, 2022.

4. Non-Dairy Frozen Dessert Market Size, Share & Trends Analysis Report By Type (Ice Cream, Sorbet, Custard, Yogurt, Gelato, 274 Sherbet, Frozen Novelties), By Distribution Channel (Food Service, Retail), By Region, And Segment Forecasts, 2022 – 2030. 275 https://www.grandviewresearch.com/industry-analysis/non-dairy-frozen-dessert-market-report. Accessed 3 August, 2022. 276

5. Allied Market Research. 2022. Vegan Ice Cream Market. https://www.alliedmarketresearch.com/vegan-ice-cream-market-A06342. 277 Accessed 3 August, 2022

6. Kades, A. Taste is king: Flexitarian consumers driving innovation for meat and dairy alternatives. 279 https://www.foodingredientsfirst.com/news/taste-is-king-flexitarian-consumers-driving-innovation-for-meat-and-dairy- 280 alternatives.html. Accessed 2 August 2022. 281

7. Fleming, A. 2017. https://www.godairyfree.org/news/generation-z-dairy-free. Accessed 4 August, 2022

17. Fleming, A. 10 Low Sugar Dairy-Free Ice Cream Brands and How They Rank. 2020. https://www.godairyfree.org/product- 301 reviews/low-sugar-dairy-free-ice-cream. Accessed August 2, 2022. 302

18. Cleveland Clinic. Sugar Substitutes & Non-Nutritive Sweeteners. 2019. https://my.clevelandclinic.org/health/ articles/15166- 303 sugar-substitutes--non-nutritive-sweeteners. Accessed 3 August, 2022 304 19. Ziesel, J. Facts about sugar and sugar substitutes. John Hopkins Medicine. https://www.hopkinsmedicine.org/health/wellness- 305 and-prevention/facts-about-sugar-and-sugar-substitutes. Accessed 4 August, 2022. 306

21. Willett, W. Coconut oil and health. 2018. Harvard Health Publishing. https://www.health.harvard.edu/staying-healthy/coconut- 310 oil. Accessed 3 August, 2022

27. Ice Cream Pints. https://www.benjerry.com/flavors/ice-cream-pints. Accessed 1 August, 2022. 321

28. All Ice Cream Products. https://www.tillamook.com/products/ice-cream/all. Accessed 1 August, 2022. 322

29. Diet and Fitness Today. Amount of vitamin B12 in ice cream. http://www.dietandfitnesstoday.com/vitamin-b12-in-ice-cream.php. 323 Accessed 4 August, 2022

Author Response

General comments

  • Overall, the paper shows a thorough look into the availability and nutrient contents on non-dairy frozen desserts in the US market. The paper is overall well written. However there is a concern on the significance of the content and the practical implications of using these findings. This is due to the focus on the nutrient content of non-dairy frozen desserts alone being too restrictive, when there are many other plant based categories to consider with, for analysis. The paper could benefit from a more in-depth discussion on the scientific/practical significance of the results. 

We appreciate the reviewer’s helpful suggestions. A sentence has been added to the abstract suggesting the implications of the findings. Additional material has been added (highlighted in yellow) to the discussion section.

  • It would have been interesting to compare the nutritional values to existing recognized profiling systems

Thank you for your observation. Unfortunately, the authors do not understand what is meant by “recognized profiling systems”. We used the standard DG low and high values for analysis of the products and focused on known nutrients that are widely recognized as conveying healthful or unhealthful properties.

  • The review is missing practical guidance: are some bases more recommended/more nutritious than others? This could be discussed more in depth in the discussion

Thanks for your helpful comments. Although Table 5 outlines those products, we have added more material in the discussion section about the more nutritious products. Most people have a personal preference based on flavor and cost as with non-dairy beverages. It can be challenging to recommend a preferred product when the different based products do not stand out in great contrast from the others as we found with the published papers on non-dairy beverages and yogurts. Nutritionally, most of the bases are very much alike because of the heavy use of sugar and coconut oil.

We have added the following comment:  All 5 brands that had the highest protein level (3 g and above) contained nuts or seeds. But their presence was not a guarantee of a high protein level, since the protein level macadamia nut based products was only 1 g and the sunflower seed based desserts had an average of 2 g protein. The frozen desserts based upon fruits (apple or pear) had the highest protein levels as the food companies had increased the protein level of adding nuts or seeds.

Added at Line 161: From Table 1 we see that coconut based products have significantly more saturated fat than 9 other based products (almond, apple/hemp, avocado, canola oil, faba protein, oats/hemp, olive oil, rice and soy).

Added at Line 163: Apple/ hemp-, avocado-, and faba bean-based products contain significantly less sugar than almond-, canola oil-, olive oil-, rice-, and sunflower-based products.

Added to the text:  Erythritol, a sugar alcohol, has 70% of the sweetness of sucrose but only 5% of the caloric value, and may cause gastrointestinal discomfort and diarrhea in large quantities. Allulose, an isomer of fructose, has antioxidant activity, and has 70% of the sweetness of sucrose. In addition, D-allulose shows anti-hyperglycemic effect in people with glucose intolerance [84]. Monk fruit (or Luo han guo) extract is up to 250 times sweeter than sugar, due mainly to the presence of compound mogroside V [82]. Monk fruit extract has no energy value, and has exhibited anti-hyperglycemic effects [81]. A triterpenoid glycoside from monk fruit has been shown to have anti-proliferative activity with the potential for treating colorectal cancer [83]. Stevia extract contains glycosides that are zero calorie sweeteners, and are 200-400 times sweeter than sucrose. All these alternative sweeteners have GRAS (general recognized as safe) status as designated by the FDA (Food and Drug Administration) in the US [81].

81.Mooradian, A.D.; Smith, M.; Tokuda, M. The role of artificial and natural sweeteners in reducing the consumption of table sugar: A narrative review. Clin Nutr ESPEN 2017 Apr;18:1-8. doi: 10.1016/j.clnesp.2017.01.004.  

  1. Bhusari, S.; Rodriguez, C.; Tarka Jr, S.M.; Kwok, D.; Pugh, G.; Gujral, J.; Tonucci, D. Comparative In vitro metabolism of purified mogrosides derived from monk fruit extracts. Regul Toxicol Pharmacol 2021;120:104856. doi: 10.1016/j.yrtph.2020.104856.

  1. Liu, C.; Dai, L.; Liu, Y.; Rong, L.; Dou, D.; Sun, Y.;Ma, L. Antiproliferative Activity of Triterpene Glycoside Nutrient from Monk Fruit in Colorectal Cancer and Throat Cancer. Nutrients 2016 Jun 13;8(6):360. doi: 10.3390/nu8060360.

[84] Hayashi, N.; Iida, T.; Yamada, T.; Okuma, K.; Takehara, I.; Yamamoto, T.; Yamada,K.; Tokuda, M. Study on the postprandial blood glucose suppression effect of D-psicose in borderline diabetes and the safety of long-term ingestion by normal human subjects. Biosci Biotechnol Biochem 2010;74(3):510-9. doi: 10.1271/bbb.90707.

What could be the reasons for such a broad variety in nutritional quality? Can the authors make some hypotheses? (brand, product base, target consumer group, …)

Thank you for your comments and questions. As seen from Table 1, most of the brands had no significant differences for each of the 8 nutrients. The large contribution of coconut oil and sugar to most of the base types masks most differences that would be expected with the diversity of base types. Eleven of the base types represented single brands only with less than 10 varieties of desserts within the brand. Hence, with such a small sample size for many of the base types statistical differences are limited.

  • Are there any claims made on these products, and if so, are those with claims of better nutritional quality?

Very few claims were made for the products beyond being plant-based and non-dairy. Some claimed to contain no sugar, and were sweetened with dates or honey or a sugar substitute. Most of these are loaded with coconut cream/coconut milk which makes them low on any recommendation list. A few mentioned non-GMO, but this is more of a marketing tool than any promise of better nutrition. Some claimed they contained no high fructose corn syrup or artificial sweeteners, but in reality none of the desserts contain either.

Materials and methods

The methodology of data collection needs to be further described. Lines 57-58 – Which or how many supermarkets were visited? What was the methodology of data collection and how was data collected to include or exclude products selection? Who collected the data?

The lead author collected 99% of the data. Apart from the manufacturers website data, all stores were from West coast supermarkets. The lead author lives in Eastern Washington and winters in Southern California. He also purchased many of the brands to sample them. Major supermarket chains were covered. Stores in Washington, Oregon, Idaho, Utah, and California (especially Los Angeles and San Bernardino counties) were visited. This data has been added to the supplemental material.

Results

Table 1 was not easy to read. The lower case letters used to indicate significance was confusing as each and every value in the table had letters affixed. It would be better to use either an * or highlight significant pairs. 

The overall p - values for each nutrient are supplied in the supplementary tables S1 & S2 and the lower case letters show which bases are significant within a nutrient category. Any bases with the same letter are not significantly different from each other. This is the standard way to display the results, and it appears complex because we are comparing so many base types.

Table 5 shows healthful nutrient levels for 1 brand for each base type, how was this 1 brand selected compared to others?

It selected itself because there were no other brands. The examples given represented single brands: Snow Monkey, Cado, Arctic Zero, Doozy Dots, Wild Good, Kind.

 Discussion

The authors mentioned dairy desserts in the introduction and conclusion as being different from plant based desserts. However, the manuscript lacked a comparison analysis of the non-dairy desserts with dairy products, which could be added to the discussion since the authors concluded plant based desserts to have high sugar and saturated fat too.  

The comparison with dairy appears in the supplementary material Table S3.

References

There is concern 12 references out of 38 were from websites that may not be scientifically credible. References questioned:

Thank you for your concerns. I also dislike to use web sources and try to keep them to a very minimum. However, there are so few scientific references available in this field it has been tough to follow my normal practice. However, I have been very selective and careful in what I have chosen and I have given reasons why the references used can be considered professionally acceptable. 

  1. Yuko, E. The History of Ice Cream, One of the World’s Oldest Desserts. March 2022. https://www.rd.com/article/who-invented- 269 ice-cream/. Accessed 2 August, 2022. 270

This article is a Readers Digest article written with the help of noted food historian Sarah Wassberg Johnson

  1. Avey, T. Explore the Delicious History of Ice Cream. July 2012. https://www.pbs.org/food/the-history-kitchen/explore-the- 271 delicious-history-of-ice-cream/. Accessed 2 August, 2022.

This article is written by Tori Avey, a credible food writer for PBS (Public Broadcasting Service) who uses 5 valid references for her article. 

  1. Non-Dairy Frozen Dessert Market Size, Share & Trends Analysis Report By Type (Ice Cream, Sorbet, Custard, Yogurt, Gelato, 274 Sherbet, Frozen Novelties), By Distribution Channel (Food Service, Retail), By Region, And Segment Forecasts, 2022 – 2030. 275 https://www.grandviewresearch.com/industry-analysis/non-dairy-frozen-dessert-market-report. Accessed 3 August, 2022. 276

This reference is a market analysis report by GVR, a recognized market analysis firm

  1. Allied Market Research. 2022. Vegan Ice Cream Market. https://www.alliedmarketresearch.com/vegan-ice-cream-market-A06342. 277 Accessed 3 August, 2022

A trustworthy source

  1. Kades, A. Taste is king: Flexitarian consumers driving innovation for meat and dairy alternatives. 279 https://www.foodingredientsfirst.com/news/taste-is-king-flexitarian-consumers-driving-innovation-for-meat-and-dairy- 280 alternatives.html. Accessed 2 August 2022. 281

Food ingredients is the leading international publisher in food ingredients and food product development

  1. Fleming, A. 2017. https://www.godairyfree.org/news/generation-z-dairy-free. Accessed 4 August, 2022
  2. Fleming, A. 10 Low Sugar Dairy-Free Ice Cream Brands and How They Rank. 2020. https://www.godairyfree.org/product- 301 reviews/low-sugar-dairy-free-ice-cream. Accessed August 2, 2022. 302

Ms Fleming has the most comprehensive write-up on non-dairy foods including frozen desserts. She updates her web site with new products all the time and works with manufacturers to have correct nutritional data.

  1. Cleveland Clinic. Sugar Substitutes & Non-Nutritive Sweeteners. 2019. https://my.clevelandclinic.org/health/ articles/15166- 303 sugar-substitutes--non-nutritive-sweeteners. Accessed 3 August, 2022 304
  2. Ziesel, J. Facts about sugar and sugar substitutes. John Hopkins Medicine. https://www.hopkinsmedicine.org/health/wellness- 305 and-prevention/facts-about-sugar-and-sugar-substitutes. Accessed 4 August, 2022. 306
  3. Willett, W. Coconut oil and health. 2018. Harvard Health Publishing. https://www.health.harvard.edu/staying-healthy/coconut- 310 oil. Accessed 3 August, 2022

References # 18,19 and 21 are websites from Cleveland Clinic, John  Hopkins University and Harvard University, all Ivy League Schools of world renown and trustworthy and scientifically credible.

  1. Ice Cream Pints. https://www.benjerry.com/flavors/ice-cream-pints. Accessed 1 August, 2022. 321
  2. All Ice Cream Products. https://www.tillamook.com/products/ice-cream/all. Accessed 1 August, 2022. 322

Ref 27 & 28, reference the company websites that must be trusted to provide accurate information about their products.

  1. Diet and Fitness Today. Amount of vitamin B12 in ice cream. http://www.dietandfitnesstoday.com/vitamin-b12-in-ice-cream.php. 323 Accessed 4 August, 2022

This is the most comprehensive analysis of a variety of ice-cream products (23 types) including cones, sandwiches and regular containers, and includes, in addition to B12, carbs, fats, protein and calories

Reviewer 2 Report

The authors explored the non-dairy products regarding nutritional value. It will be great interest for the readers of Nutrients and provide great insights to researchers, educators and product formulators. I only have a few questions/suggestions:

1. The readers may interested in details of the these 358 nutrition labels and would practice to replicate the results. 

2. The readers may also interested in the R codes for analyzing the data. 

3. Supporting information cannot be downloaded from my side

Author Response

Comments and Suggestions for Authors

The authors explored the non-dairy products regarding nutritional value. It will be great interest for the readers of Nutrients and provide great insights to researchers, educators and product formulators. I only have a few questions/suggestions:

Thank you for your affirming words.

  1. The readers may interested in details of these 358 nutrition labels and would practice to replicate the results.

Thanks for your concerns. These are retrievable from the food company websites if anyone wishes to duplicate the work. 

  1. The readers may also interested in the R codes for analyzing the data. 

Anyone can email the authors for the R codes once they have re-created the original data set.

  1. Supporting information cannot be downloaded from my side

Sorry for that problem. Perhaps this was due to the .docx format. We have attached it for you as pdf file so it is more easily accesible.

Reviewer 3 Report

The article presents the composition of several non diary frozen deserts, regarding main nutrients of interest, clasifying them against recomanded daily intakes in the USA. Results are interesting, especially due to the late public interest in this type of products. High levels of sugars and saturated fats were noticed, as well as insufficient protein and some vitamins, that are generally present in diary based deserts. 

Tabels are clear, with the exception of instances when lower cases are used. Please, provide clearer explanations for them. (a, b...)

The discussion chapter is not adequate, mainly presenting data we can read ourselves from tabels. More detailed analyzis regarding the importance and impact of the results on consumers' health is needed, as well as the need to reformulate these products, in order to ameliorate their composition. Ditto for conclusions. E.g, results indicate the need for better recipes of these products, with lower sugar and SFA levels. A need for fortification with certain vitamins and minerals might be highlighted.

References are adequate. The subject is moderately interesting, however, the article needs substantial improvements.

Author Response

The article presents the composition of several non-diary frozen deserts, regarding main nutrients of interest, clasifying them against recomanded daily intakes in the USA. Results are interesting, especially due to the late public interest in this type of products. High levels of sugars and saturated fats were noticed, as well as insufficient protein and some vitamins, that are generally present in diary based deserts.

Thank you for helping us to make the manuscript better. 

Tabels are clear, with the exception of instances when lower cases are used. Please, provide clearer explanations for them. (a, b...)

Thank you for your concerns. The overall p - values for each nutrient are supplied in the supplementary tables S1 & S2 and the lower case letters show which bases are significant within a nutrient category. Any bases with the same letter are not significantly different from each other. Significant post-hoc pairwise comparisons are shown by different lowercase letters. This is the standard way to display the results, and it appears complex because we are comparing so many base types with similar medians.

We have added details in lines 94, 111, and 149 – 151 to make this explanation clearer.

The discussion chapter is not adequate, mainly presenting data we can read ourselves from tabels. More detailed analyzis regarding the importance and impact of the results on consumers' health is needed, as well as the need to reformulate these products, in order to ameliorate their composition. Ditto for conclusions. E.g, results indicate the need for better recipes of these products, with lower sugar and SFA levels. A need for fortification with certain vitamins and minerals might be highlighted.

We appreciate your concern to improve the paper and we have tried to do this. Added text as suggested:

Manufacturers need to provide healthier non-dairy frozen dessert options for those consumers who wish to consume more than just a tasty, sweet dessert. To begin with, this could be achieved using lower levels of coconut oil and sugar. Food manufacturers should also consider fortifying such desserts with calcium to levels similar to those found in regular dairy ice-cream.

We have added the following information on Lines 255:

The type of base did not affect calorie, fat, carbohydrate, sugar or protein content (Table 3). This may be due to the small sample size, and variable serving size.

Additions to the manuscript are highlighted in yellow

Research reported earlier by the authors revealed that the non-dairy beverages and yogurts showed considerable variation in nutritional content depending upon the various base types. However, the non-dairy frozen desserts generally lacked that diversity depending upon base type.

References are adequate. The subject is moderately interesting, however, the article needs substantial improvements.

Thank you for your concerns. The highlighted text in yellow shows the additions we have made to the document to improve it according to all of the reviewers comments. We hope you are pleased with the improvements we have made at your suggestion.

Round 2

Reviewer 1 Report

The authors have addressed all major points listed in the review. 

Reviewer 3 Report

The paper has been substantially improved, thank you for taking the time to improve it.